# Radioactive Seed Localization for Nonpalpable Breast Lesions: Systematic Review and Meta-Analysis

**DOI:** 10.3390/diagnostics14040441

**Published:** 2024-02-17

**Authors:** Hortência H. J. Ferreira, Carla Daruich de Souza, Lorena Pozzo, Martha S. Ribeiro, Maria Elisa C. M. Rostelato

**Affiliations:** Nuclear and Energy Research Institute (IPEN/CNEN—SP), University of São Paulo (USP), Av. Professor Lineu Prestes 2242, São Paulo 05508-000, SP, Brazil; hortenciaferreira.radiologia@gmail.com (H.H.J.F.); lorena.pozzo@ipen.br (L.P.); marthasr@usp.br (M.S.R.); elisaros@ipen.br (M.E.C.M.R.)

**Keywords:** breast-conserving surgery, intraoperative localization, radioactive seed localization, systematic review, meta-analysis

## Abstract

Background: This study is a systematic review with meta-analysis comparing radioactive seed localization (RSL) versus radio-guided occult lesion localization (ROLL) and wire-guided localization (WGL) for patients with impalpable breast cancer undergoing breast-conserving surgery and evaluating efficacy, safety, and logistical outcomes. The protocol is registered in PROSPERO with the number CRD42022299726. Methods: A search was conducted in the Embase, Lilacs, Pubmed, Scielo, Web of Science, and clinicaltrials.gov databases, in addition to a manual search in the reference list of relevant articles, for randomized clinical trials and cohort studies. Studies selected were submitted to their own data extraction forms and risk of bias analysis according to the ROB 2 and ROBINS 1 tools. A meta-analysis was performed, considering the random effect model, calculating the relative risk or the mean difference for dichotomous or continuous data, respectively. The quality of the evidence generated was analyzed by outcome according to the GRADE tool. Overall, 46 articles met the inclusion criteria and were included in this systematic review; of these, 4 studies compared RSL and ROLL with a population of 1550 women, and 43 compared RSL and WGL with a population of 19,820 women. Results: The results showed that RSL is a superior method to WGL in terms of surgical efficiency in the impalpable breast lesions’ intraoperative localization, and it is at least equivalent to ROLL. Regarding security, RSL obtained results equivalent to the already established technique, the WGL. In addition to presenting promising results, RSL has been proven to be superior to WGL and ROLL technologies.

## 1. Introduction

Breast cancer is classified as a malignant, genetic, and multifactorial disease. It usually starts in the lobular duct from cellular mutations that develop into DNA damage [1]. This is a complex and gradual process that is related to endocrine, nutritional, and environmental factors. According with the World Health Organization, 685,000 people worldwide died in 2020 as a result of breast cancer, which affected 2.3 million women. The most common cancer in the globe as of the end of 2020 was breast cancer, which had been diagnosed in 7.8 million women in the previous five years [2]. 

Recommendations for breast cancer detection include participation in a mammographic screening program, that is, a periodic mammography in women who do not present signs or symptoms of the disease [3], which facilitates an early diagnosis, when the cancer is in its early stages. As a result of such programs, currently, one-third of diagnosed lesions are non-palpable [3,4].

The main treatment for breast cancer is surgery, which can be associated with radiotherapy or systemic therapies. Currently, the first choice of treatment for patients with early breast cancer is breast-conserving surgery, a method that prioritizes tumor removal with a safety margin, keeping as much breast tissue as possible [5]. The therapeutic result of breast-conserving surgery does not differ from that of radical surgery [6]. Its benefits are associated with smaller excised tissue volumes, which contribute to the aesthetic needs of patients, better prognosis, and life quality preservation. As the breast is partially excised, without dissection of axillary lymph nodes, it results in conservation of the movement of the affected arm, reducing the lymphedema risk rate [6,7].

Conservative surgery of nonpalpable breast lesions requires an intraoperative localization method. The purpose of this method is to ensure accurate identification of the lesion site during surgery and guide the surgeon in the complete removal of the tumor, sparing tissues and structures not infiltrated by tumor cells [8].

Wire-guided localization (WGL) is currently the most used method [8]. Radio-guided localization techniques, radio-guided occult lesion localization (ROLL), and radioactive seed localization (RSL) are promising alternatives for use in the localization and excision of these lesions [9].

The WGL technique consists of introducing a wire through a guide needle into the lesion area. After stabilizing the wire position, the needle is withdrawn, and the wire projects out of the breast, which must be secured with adhesive tape to keep it in place until surgery. The marking can be guided by ultrasound or stereotaxic mammography. After the procedure, mammography is mandatory to confirm the wire’s location. In this method, the surgical procedure is performed on the same day as the wire implantation to avoid dislocations and infections [10].

The ROLL technique consists of the use of a radiotracer, where about 0.2 to 0.5 mL of 99m-technetium-labeled colloidal albumin (^99m^Tc-MAA) is injected directly into the breast [9], under ultrasound or stereotactic mammography guidance. As technetium is a gamma-type radiation emitter [11], a gamma detector device is needed to identify the marker during surgery [12]. It is possible to perform ROLL in conjunction with sentinel lymph node biopsy (SLNB) using the same radioactive tracer. Because the half-life of technetium-99m is short (6 h), surgery should be performed within 24 h after the lesion is marked [13].

The RSL technique consists of implanting one or more iodine-125 radioactive seeds in the lesion area, under ultrasound or stereotaxic mammography guidance [13]. After the procedure, mammography is performed to confirm the seed location [14]. The use of a gamma detector device, such as a gamma probe, is necessary during surgery to identify the seed location [12,13,14].

The radioactive seed consists of a titanium capsule with millimeter dimensions, and its internal structure contains a radio-opaque marker radiolabeled with iodine-125, varying according to the manufacturer [15,16]. Some have the radioactive material distributed in resin spheres or ceramic matrix, others on a silver wire [17,18].

Iodine-125 has a half-life of approximately 60 days [11], emitting gamma radiation with an average energy of 29 KeV [9], and the activity per seed is between 2 and 13 MBq [15]. Thus, the iodine-125’s physical characteristics permit the surgery to be performed within 2 months, on average, after implantation [12]. In addition, seed insertion can be performed before the initiation of neoadjuvant chemotherapy, and patients can avoid a new procedure the day of the surgery [19,20]. The gamma probe can discriminate energy, so SLNB can be performed in the same surgical procedure because the iodine-125 seed emits ≈29 KeV photons and technetium-99m emits ≈140KeV photons [9,12,13].

The aim of this study was to develop a systematic review with meta-analysis to evaluate the technique of impalpable breast lesions’ intraoperative localization with RSL with regard to its efficacy, safety, and logistics compared to WGL and/or ROLL.

## 2. Materials and Methods

The systematic review protocol is registered in PROSPERO with the number CRD42022299726. In Appendix A the search strategy, eligibility criteria of studies for review, summary of the results, and observations are presented.

### 2.1. Research Methods to Identify and Select Studies

Databases: The searches were performed in the following databases: Embase, LILACS, PubMed, SciELO, Web of Science, and clinicaltrials.gov. In addition, a manual search was performed in reference lists. Studies published in languages other than English, Portuguese, and Spanish were excluded. The search was performed to include studies published during the researched period. The search was performed again before the final analysis to include possible recent eligible studies.Search strategy: The identification scope of relevant studies was as wide as possible according to the inclusion criteria. Based on this, the search strategy was to write with controlled and free vocabulary, respecting the syntax of each database. Exclusion filters, including the publication year, were not used. The database search strategy can be found in the Appendix A.Outcomes: Positive surgical margins, intraoperative re-excision, reoperation, recurrence, failed identification of SLNB, specimen volume, complications, patient assessment of intervention, medical team assessment of intervention, localization time, operative time, and time between site and surgery.Two-step triage: Studies identified through the search strategy were exported to Mendeley [21] for managing references and removing duplicates. The study selection was performed through a two-step triage process, performed independently by two of the authors (Ferreira, Rostelato). Any discrepancies regarding eligibility were resolved by discussion among all authors. In the first step, the triage was performed by reading the title and abstract, to select all studies that potentially met the inclusion criteria. In the second step, the full text of all studies selected in step one was analyzed to confirm eligibility or exclusion from the study. The research results, including the studies’ selection and justified exclusion, were organized in a selection flowchart, according to the guidelines presented in “Preferred Reporting Items for Systematic review and Meta-analysis Protocols” (PRISMA-P) [21].

### 2.2. Eligibility Criteria of Studies for Review

PopulationInclusion: women, of any age over eighteen years old, diagnosed with a nonpalpable breast lesion, eligible for surgical intervention. Exclusion: age under 18, pregnancy, men. 

InterventionInclusion: RSL with iodine-125 for nonpalpable breast lesion conservative surgery. Exclusion: RSL for regions other than the breast. 

ComparatorInclusion: WGL and ROLL for nonpalpable breast lesion conservative surgery. Exclusion: WGL and ROLL for regions other than the breast. 

Study designInclusion: randomized clinical trial and cohort study with more than 10 patients, comparing RSL against ROLL and/or WGL. Exclusion: reviews, letters, abstracts, comments, in vitro assays

### 2.3. Data Analysis

Data extraction: Data from all studies included were collected independently by two of the authors (Ferreira, Rostelato). Any discrepancies regarding the collected data were resolved by discussion among all authors. A standard form created by the authors was used in this process to collect the following data: study information (authors, title, year of publication, country, type of study); population characteristics (number of participants, age, clinical and pathological information about the lesion); methodology (assessed interventions, procedure description, assessed outcomes); results (description of results by outcome).Risk of bias assessment: The risk of bias assessment was performed independently by two authors (Ferreira, Rostelato). Any discrepancies regarding the critical assessment were resolved by discussion among all authors. The evaluation was performed for all included studies, using its specific tool for each study design. The risk of bias assessment results was illustrated in graphs with the most important points that could add bias to the review, using the Robvis tool [22]. Risk of bias assessment tools are structured with fixed bias domains, focusing on different aspects of study design, conduct, and reporting. Within each domain, there is a series of questions that aim to obtain information about study characteristics that are relevant to the risk of bias.
○Randomized clinical trials were evaluated according to ROB 2, which is the Cochrane risk of bias tool for randomized trials [23]. The domains evaluated with this tool are random generation and allocation; blinding of participants, professionals, and outcome evaluators; incomplete outcomes; outcome measurement; and selective outcome reporting. ○Observational studies were evaluated according to ROBINS I [24], which is the Cochrane risk of bias tool for evaluating non-randomized studies of interventions. With this tool, the domains evaluated were confusion; study participants selection; intervention classification; deviations from the intended interventions; incomplete outcomes; outcome measurement; and selective outcome reporting.
Measures of treatment effectiveness: A combined estimate for effect size and 95% confidence interval (CI = ) was calculated by combining all studies to determine the likely beneficial therapeutic outcome between treatment approaches. Dichotomous data were measured using the relative risk (RR) for SLNB identification failure rate, complication rate, positive resection margin rate, intraoperative re-excision rate, reoperation rate, recurrence rate, and intervention evaluation by the patient. Mean difference (MD) was calculated to measure continuous data such as localization time, time between localization and surgery, operating time, and specimen volume. The relative risk (RR) or odds ratio is the ratio of the probability that an individual in the exposed group will develop the condition studied to the probability that an individual in the comparator group will develop the same condition.Lost data: This was an available case analysis, where data were analyzed for known cases only. To solve the missing data problem, all data were obtained whenever possible by contacting the correspondent author. In cases where data were found to be randomly lost, only existing data were analyzed. All known missing data occurrences will be presented and explored in the Section 4.Assessment of heterogeneity: A statistical heterogeneity test was conducted to detect any differences in effects between the studies. This was performed by reviewing the confidence intervals (CI) for each of the individual studies in the systematic review to assess overlap; and formally conducting a statistical test for heterogeneity using the I^2^ statistical test. An I^2^ greater than 50% indicates substantial heterogeneity, and above 75% considerable heterogeneity. The analyzed results were organized into forest plots using the Review Manager software, version 5.4, Cochrane Library, Hoboken, NJ, USA [25].Assessment of publication bias: To analyze the issue of publication bias, risk assessment forms were used to determine the level and type of risk of bias. This information was then entered into the Review Manager 5 software, where the results were summarized in funnel plots, except for outcomes that had less than 10 studies grouped together, because the test’s power is too low to determine symmetry.Data synthesis: The random effects model was chosen to group the data by outcome, using the Review Manager 5 software. Results were reported in forest plots presenting the effect estimate. For quantitative data that were not considered appropriate for grouping due to heterogeneity or missing data, the synthesis was performed qualitatively with a narrative description.Evidence quality: The evidence quality was evaluated for each of the outcomes independently according to the orientations present in the GRADE tool, which is a system for grading evidence and strength of recommendations [26]. The level of evidence represents the confidence level in estimating the effects presented in support of a particular recommendation and can be classified into four levels: high, moderate, low, and very low. The factors responsible for classifying the evidence are related to methodological quality, inconsistency, imprecision, indirect evidence, and publication bias.

## 3. Results

The database search, performed until 30 November 2023, and the manual reference list search retrieved 4499 records without duplicates, of which 4218 were excluded after screening by title and abstract. Finally, we excluded a further 232 articles in the full-text screening. In total, 46 articles met the inclusion criteria and were included in this systematic review, comprising 10 randomized clinical trials and 36 observational studies. For the study selection flowchat, see Figure 1 [21].

### 3.1. RSL and ROLL 

#### 3.1.1. Comparative Analysis by Outcome

Four observational studies [19,27,28,29] compared RSL and ROLL. The data synthesis from the four studies comprised a population of 1550 women with nonpalpable breast cancer, with ages ranging from 28 to 91 years old, undergoing conservative surgery with RSL or ROLL. Patients with bracketing localization, when two or more markers are used per breast, and/or undergoing neoadjuvant chemotherapy were also included. The general description of the patients’ characteristics is presented in Table 1.

Outcomes of intraoperative re-excision, failed identification of SLNB, patient assessment of intervention, medical team assessment of intervention, localization time, and operative time were not reported by the studies included in this systematic review. Outcomes of specimen volume and complications were only reported by one study, and therefore data synthesis was not performed. The outcomes of positive surgical margins, reoperation, recurrence, and time between site and surgery are presented below.

#### 3.1.2. Positive Surgical Margins

Positive surgical margin rates were reported by three studies [19,27,29], with a low heterogeneity (I^2^ = 39%). The positive surgical margins risk in conservative surgery with RSL was lower than with ROLL, but this value was not statistically relevant (RR = 0.83, 95% CI = 0.50–1.39; 763 patients; three studies; see Figure 2).

#### 3.1.3. Reoperation

Reoperation rates were reported by four studies [19,27,28,29], with a low heterogeneity (I^2^ = 17%). The reoperation risk after conservative surgery with ROLL was lower than with RSL, but this value was not statistically relevant (RR = 1.14, 95% CI = 0.75–1.74; 1550 patients; four studies; see Figure 3).

#### 3.1.4. Recurrence 

Recurrence rates—local recurrence, regional recurrence, and distant metastasis—were reported by three studies [27,28,29], without heterogeneity (I^2^ = 0%). The recurrence risk after conservative surgery with RSL was lower than with ROLL (RR = 0.50, 95% CI = 0.29–0.87; 939 patients; three studies; see Figure 4).

#### 3.1.5. Time between Localization and Surgery 

The time interval between localization and surgery was reported by three studies [19,27,28], totaling 1301 patients (RSL = 625; ROLL = 676); however, due to missing data, it was not possible to perform the statistical synthesis. The time interval between localization and surgery was longer with RSL, 1 day–31 weeks, than with ROLL, 0–1 day.

#### 3.1.6. Risk of Bias Assessment

The risk of bias in the included studies that evaluated RSL x ROLL was considered moderate to low. Figure 5 presents the general evaluation result of these studies, according to the ROBINS I tool, 2016 version, Cochrane Library, Hoboken, NJ, USA, as they are non-randomized studies.

The main limitations that may introduce bias in the review are related to the domains of result measurement and confounding. For the studies by Donker [27], Niinikoski [28], and Theunissen [29], the methods for evaluating the results were not comparable between the intervention groups for the outcome of recurrence, as the follow-up period was different between the studied groups.

The study by Van der Noorda [19] showed the potential for confounding the intervention effect at baseline because the clinical and pathological characteristics of patients are different between the two groups studied, which may influence treatment decisions, and there is no information on whether the authors used a method of analysis that controlled for important confounding domains. However, the surgical efficiency analyses were performed by subgroup so that the confounding domain did not interfere with these outcomes.

#### 3.1.7. Evidence Quality Assessment

For the outcomes of surgical margins, reoperation, and recurrence, the quality of evidence was classified as high. There is no relevant presence of inconsistency (I^2^ = 39%, 17% and 0%) or imprecision (CI = 0.5–1.39; 0.75–1.74; 0.29–0.87) for surgical margins, reoperation, and recurrence, respectively. Indirect evidence was not used for any of the outcomes. The risk of bias in the individual studies is low for all outcomes except recurrence, which had a moderate risk of bias. In contrast, the effect magnitude presented on the recurrence outcome is high (RR ≤ 0.5). Publication bias was not assessed due to the small number of studies grouped for each outcome.

The assessment of evidence quality for the time between localization and surgery outcome is inconclusive due to the absence of statistical synthesis. It is not possible to evaluate the inconsistency, imprecision, or publication bias of the results. However, it is noteworthy that the risk of bias in the included studies concerning this outcome is low, and there is no significant presence of indirect evidence.

### 3.2. RSL and WGL

#### 3.2.1. Comparative Analysis by Outcome

Overall, 43 studies compared RSL and WGL, including 7 randomized controlled trials [30,31,32,33,34,35,36,37], and 32 observational studies [29,38,39,40,41,42,43,44,45,46,47,48,49,50,51,52,53,54,55,56,57,58,59,60,61,62,63,64,65,66,67,68,69]. The data synthesis from the 39 studies comprised a population of 19,820 women with nonpalpable breast lesions, with ages ranging from 20 to 92 years old, undergoing conservative surgery with RSL or WGL. Patients with bracketing localization and/or undergoing neoadjuvant chemotherapy were also included. The general description of the patients’ characteristics is presented in Table 2.

#### 3.2.2. Positive Surgical Margins 

Positive surgical margin rates were reported by 33 studies [29,30,31,33,34,35,38,39,40,41,42,44,45,46,47,48,49,50,51,52,53,54,55,56,57,58,60,64,65,67,68,69,70], with a low heterogeneity (I^2^ = 36%). The positive surgical margins risk in conservative surgery with RSL was lower than with WGL (RR = 0.78, 95% CI = 0.70–0.87; 15,610 patients; 33 studies; see Figure 6). The narrowest part of the funnel plot is where the most accurate studies are located. This indicates that there is a predominance of studies with higher prevalence, with low risk of publication bias.

#### 3.2.3. Intraoperative Re-Excision 

Intraoperative re-excision rates were reported by 15 studies [31,34,35,40,43,45,50,62,63,64,65,66,69,70], with a high heterogeneity (I^2^ = 95%). The intraoperative re-excision risk in conservative surgery with RSL was higher than with WGL, but this value was not statistically relevant (RR = 1.02, 95% CI = 0.75–1.38; 5623 patients; 15 studies; analysis Figure 7). In the funnel plot, there was a predominance of less accurate studies, symmetrically located in the widest region of the funnel, indicating possible publication bias.

#### 3.2.4. Reoperation 

Reoperation rates were reported by 27 studies [29,35,38,39,40,41,42,44,45,46,47,48,49,50,51,52,53,54,55,56,57,60,61,62,63,64,65,66,67,69,70], with a relevant heterogeneity (I^2^ = 61%). The reoperation risk after conservative surgery with RSL was lower than with WGL (RR = 0.71, 95% CI = 0.61–0.84; 13884 patients; 27 studies; see Figure 8). The narrowest part of the funnel plot, where the most accurate studies are located, has a predominance of studies with higher prevalence, signifying a low risk of publication bias.

#### 3.2.5. Recurrence 

Recurrence rates—local recurrence, regional recurrence, and distant metastasis—were reported by three studies [29,32,39], with no heterogeneity (I^2^ = 0%). The recurrence risk after conservative surgery with RSL was lower than with WGL (RR = 0.41, 95% CI = 0.19–0.86; 1525 patients; three studies; see Figure 9).

#### 3.2.6. Sentinel Lymph Node Biopsy Failure Identification Rate

SLNB failure rates were reported by six studies [30,33,34,45,48,70], with no heterogeneity (I^2^ = 0%). The SLNB failure risk associated with conservative surgery with RSL and WGL was equivalent (RR = 1.00, 95% CI = 0.35–2.87; 1318 patients; six studies; see Figure 10).

#### 3.2.7. Complications 

The complications rates related to marker implantation were reported by 25 studies [30,33,34,35,41,42,43,44,45,46,47,48,49,50,51,52,55,57,60,61,64,67,68,69,70], with a relevant heterogeneity (I^2^ = 57%). The complications risk in conservative surgery with WGL was lower than with RSL, but this value was not statistically relevant (RR = 1.12, 95% CI = 0.66–1.78; 8672 patients; 25 studies; see Figure 11).

For RSL, complications involved: wrong initial implant of seeds (n = 45); marker placed more than 5 mm from the lesion (n = 17); seed displacement (n = 14); seed dislodgement during surgery (n = 10); failure to remove the seed in the first specimen (n = 10); seed loss (n = 1); seed transection (n = 1); and vasovagal reaction (n = 1).

For WGL, complications involved: wrong initial implant of the wire (n = 31); marker placed more than 5 mm from the lesion (n = 9); accidental marking (n = 1); wire displacement (n = 44); wire dislodgement during surgery (n = 6); failure to remove the wire in the first specimen (n = 7); wire breakage (n = 2); vasovagal reaction (n = 12).

In the funnel plot, there was a predominance of less precise studies, which are asymmetrically located in the widest region of the funnel, indicating possible publication bias.

Post-operative complication rates were reported by 13 studies [29,31,32,41,42,44,48,51,60,61,64,65,66], without heterogeneity (I^2^ = 0%). The post-operative complications risk after conservative surgery with WGL was lower than with RSL, but this value was not statistically relevant (RR = 1.17, 95% CI = 0.90–1.52; 5525 patients; 13 studies; see Figure 12). Post-operative complications include hematoma, infection, inflammation, seroma, wound rupture, lymphedema, and deep vein thrombosis. The funnel plot shows a predominance of studies with higher prevalence, which indicates a low risk of publication bias.

#### 3.2.8. Time between Localization and Surgery 

The time interval between localization and surgery was reported by 25 studies [30,33,34,39,40,41,42,43,44,45,47,48,49,50,52,53,55,56,60,61,64,66,67,69,70], totaling 10455 patients (RSL = 4895; WGL = 5560). However, due to missing data it was not possible to perform statistical synthesis. The time interval between localization and surgery was longer with RSL, 0–47 days, than with WGL, 0–1 day.

#### 3.2.9. Risk of Bias Assessment

The risk of bias in the included randomized trials that evaluated RSL x WGL was considered moderate to low. Figure 13 presents the general result of the evaluation of these studies, according to the ROB 2 tool. Similarly, the risk of bias in the included non-randomized studies that evaluated RSL X WGL was considered moderate to low, except for one study that had a high risk of bias [42]. Figure 14 presents the general evaluation result of these studies, according to the ROBINS I tool.

The main limitations of randomized trials that may introduce bias in the review are related to the domains of incomplete outcomes and other risks of bias such as positive/negative memory cognitive. In Gray’s study [33], five participants were excluded due to missing data. In Bloomquist’s study [30], more participants in the WGL group completed the technology assessment compared to the RSL group. In Parvez’s study [37], there was a loss of participants in the follow-up period, in addition to a loss of data from the evaluation’s initial year.

The studies by Bloomquist [30], Chagpar [31], Fung [32], Gray [33], and Parvez [37] present a sampling bias risk due to the small sample size of the study or a specific group. The Bloomquist study [30] presented a risk of memory bias because the groups were not interviewed in the same period after the procedure for the intervention evaluation outcome. Ong’s study [36] presented a risk of positive/negative memory cognitive bias for the technology assessment outcome, as the potential benefits of RSL over WGL were detailed in the study’s consent information.

The main limitations of non-randomized studies that may introduce bias in the review are related to the domains of confounding, intervention classification, incomplete outcomes, result measurement, and selective outcomes report. The studies by Horwood [46], Jumaa [49], Parvez [55], and Silva [63] did not present information on the intervention classification.

The studies by Aljohani [39], Chiu [42], Diego [43], Horwood [46], Hout [47], Hughes [48], Jumaa [49], Murphy [53], Parvez [55], Pieri [56], Rao [57], Romanoff [59], Sanchez [60], Saphier [61], Srour [64], Srour [65], Stelle [66], and Zénzola [68] showed confusion about the intervention effect on the baseline as the patient’s clinical and pathological characteristics are different between the two study groups, being potential factors to influence treatment decisions, and the authors did not use a regressive analysis method that controlled for this domain.

Chiu’s study [42] did not present information on the intervention classification, nor on the methodology used for data collection and result evaluation. Result data were not available for all participants, and there is no information on the reason or proportion of missing data. Thus, there is insufficient information to assess whether the reported effect estimate was selected based on the results of multiple measurements or different subgroups.

In the studies by Gray [45] and Hughes [48], data for the technology assessment outcome were not available for all participants. The missing data are disproportionate between interventions, and the results were not robust to the missing data. In addition, the result assessment methods were not comparable between the intervention groups for this outcome, as the self-assessment questionnaire was not performed by all patients in the WGL group.

In Romanoff’s study [59], result data were not available for all participants. The missing data are disproportionate between interventions, and the outcomes were not robust to the missing data. In Theunissen’s study [29], the methods for evaluating the results were not comparable between the intervention groups for the recurrence outcome, as the follow-up period was different between the studied groups.

Because the high risk of bias and missing data between studies, the outcomes of specimen volume, operative time, localization time, and intervention evaluation by the patient were excluded from the review in the comparative analysis between RSL and WGL.

#### 3.2.10. Evidence Quality Assessment

For the outcomes of surgical margins, reoperation, recurrence, marker-related complications, and post-operative complications, the evidence quality was classified as high. There is no relevant presence of inconsistency (I^2^ = 36%, 61%, 0%, 58%, 0%) or imprecision (CI = 0.7–0.87; 0.61–0.84; 0.19–0.86; 0.61–1.78; 0.90–1.52) for surgical margins, reoperation, recurrence, marker-related complications, and postoperative complications, respectively. Indirect evidence was not used for any of the outcomes. The risk of bias in the individual studies is low for all outcomes except recurrence, which had a moderate risk of bias. In contrast, the effect magnitude presented on the recurrence outcome is high (RR ≤ 0.5). The funnel plot of these outcomes was symmetrical, indicating a low risk of publication bias, except for the recurrence outcome, which was not evaluated due to the low number of summarized studies.

For the outcomes of intraoperative re-excision and SLNB identification failed, the evidence quality was moderate. No indirect evidence was used for these outcomes, and the risk of bias from individual studies was low. Specifically, the intraoperative re-excision outcome did not present a relevant imprecision (CI = 0.75–1.38) or publication bias in the funnel plot, however, it presented a relevant heterogeneity (I^2^ 95%). The SLNB identification failure outcome did not present heterogeneity (I^2^ 0%); however, it presented a relevant imprecision (CI = 0.35–2.87). Publication bias for this outcome was not assessed due to the low number of summarized studies.

The evidence quality assessment for the outcome of time between localization and surgery is uncertain because statistical synthesis was not performed. It is not feasible to assess the inconsistency, imprecision, or publication bias of the results. Overall, the risk of bias for this outcome is low, and there is no relevant presence of indirect evidence.

## 4. Discussion

### 4.1. Key Findings and Explanations 

The outcomes of surgical margins, intraoperative re-excision, reoperation, recurrence, SLNB failure, and specimen volume are related to the intervention’s effectiveness. The outcome of complications addresses safety, and the time between localization and surgery is related to the service’s logistical organization. 

The success of breast-conserving surgery depends on the tumor’s complete removal. Thus, intraoperative re-excision during the first surgery can be performed to excise possible positive margins identified in the specimen preparation. This is important because surgery completion with negative surgical margins decreases the risk of reoperation and disease recurrence. Conservative surgery together with SLNB is a common procedure in surgical practice. In this scenario, it is essential that one intervention does not interfere with the result of the other. 

Regarding surgical efficacy, the comparison between RSL and ROLL results did not demonstrate statistically significant superiority of one technology over the other for the reported outcomes of positive surgical margins, reoperation, and recurrence. This suggests that RSL is at least equivalent in surgical efficacy to ROLL technology. Considering that there is strong confidence that the true effect is close to the estimated one, given the high evidence quality, it is unlikely that further work will modify the confidence in the effect estimate.

Regarding safety, there is no evidence to support one technique over another with respect to this outcome. Regarding the organization of services, RSL proved to be superior to ROLL, with a longer time interval between localization and surgery.

In the comparative analysis between RSL and WGL, regarding surgical efficacy, RSL, despite having higher intraoperative re-excision rates, obtained lower rates of positive surgical margins, reoperation, recurrence. Regarding the identification failure rate in the BLS, the two technologies demonstrated equivalent efficiency. These results suggest that RSL is superior to WGL in terms of surgical efficiency. Considering that there is strong confidence that the true effect is close to the estimated one, given the high evidence quality, it is unlikely that further work will modify the confidence in the effect estimate. 

Regarding safety, WGL obtained superior, but not statistically significant, results compared to RSL in the outcomes of complications. This suggests that RSL is at least equivalent in safety to WGL technology and showed a longer time interval between localization and surgery.

The longer interval time between the localization with RSL and the surgery results in significant efficiency gains for the radiology team, due to the ability to perform all the procedures of the week in a single list, and for the surgical team, as the patient undergoing RSL can be the first of the day with no delays associated with waiting for the implant to be performed on the same day. Furthermore, in this scenario, the radioactive seed can be implanted before the initiation of neoadjuvant chemotherapy, which represents safety for patients who achieve a complete pathological response, because the seed will continue to mark the tumor site for further surgery even after tumor regression. In addition, patients who receive RSL before neoadjuvant chemotherapy avoid another invasive intraoperative localization device placement procedure before surgery.

### 4.2. Strengths and Limitations 

The main limitations of this review are related to safety outcomes, regarding the comparison between RSL and ROLL, as no primary studies were found in the literature that would serve as a basis for this analysis. Regarding the comparison between RSL and WGL, the main limitations of the review are related to the outcomes of specimen volume, operative time, localization time, and intervention evaluation by the patient, which showed low evidence quality and were excluded from the review. In addition, the outcome of the intervention evaluation by the medical team was not evaluated, as no primary studies were found in the literature that reported this result. 

These limitations demonstrate gaps in the literature that may serve as clinical questions for future research. This gap presents the need for new primary studies that address the comparison between RSL and ROLL with regard to safety and organization of services. Furthermore, even in the surgical efficiency outcomes, the sample available in the literature is small, requiring more randomized clinical trials and well-defined cohort studies to address these outcomes, so that the statistical analyses are more robust and significant. The same occurs when comparing RSL and WGL in terms of safety and organization of services. The available sample of primary studies available in the literature is small and heterogeneous.

### 4.3. Comparison with Similar Researches 

The results presented in this study on the technology’s effectiveness corroborate the systematic reviews performed by Chan [71] and Moreira [72]. In Chan’s study [71], the risk of positive surgical margins (RR = 0.67, 95% CI = 0.43–1.06; 366 participants) and reoperation (RR = 0.80, 95% CI = 0.48–1.32; 305 participants) with RSL was lower than with WGL, but this value was not statistically relevant, probably due to the small sample present in the study. In Moreira’s study [72], the positive surgical margins risk with RSL was lower than with WGL (RR = 0.84, 95% CI = 0.71–0.99; 8702 participants). RSL also showed an advantage over ROLL in this outcome, but it was not significant (RR = 1.22, 95% CI = 0.83–1.81; 609 participants). Regarding reoperation rates, RSL had lower rates than WGL (RR = 0.73, 95% CI = 0.58–0.92; 7634 participants) and showed a non-significant advantage over ROLL (RR = 1.07, 95% CI = 0.63–1.83; 609 participants). A non-significant advantage of RSL over WGL was demonstrated in terms of recurrence rates (RR = 0.55, 95% CI = 0.25–1.17; 1526 participants).

In both studies by Chan [71] and Moreira [72], RSL proved to be superior in surgical efficiency outcomes when compared to ROLL and WGL. However, this result was not statistically relevant, which can be explained due to the smaller sample of primary studies incorporated in these systematic reviews. The systematic review developed in this research is updated with a more robust sample, presenting statistically significant comparative data that prove the superiority of RSL compared to WGL. Regarding the RSL x ROLL comparison, our sample was not yet large enough to demonstrate a relevant statistical difference between the two technologies.

### 4.4. Implications and Actions Needed

This systematic review finds RSL to be superior to other techniques. An implementation protocol is needed to aid physicians and hospitals to adopt it in their clinical practice.

Initiating the implementation of the RSL involves several crucial steps. These include the formation of a procedural committee, providing education to staff members (covering areas such as radiology, surgery, pathology, radiological safety, breast imaging, and nuclear medicine), establishing a radiological safety committee, and gaining approval from a dedicated committee to investigate and adhere to federal and state-specific nuclear regulatory standards. Additionally, creating an inventory of materials and instrumentation associated with the technique, implementing a seed-tracking system, and developing emergency recovery procedures are indispensable components.

The RSL program necessitates the formulation of an institution-specific standard operating procedure protocol. This protocol is designed to outline professional practices related to radiation safety aspects, the requisition and storage of seeds, obtaining patient consent, conducting radiological procedures for placing breast localization devices, carrying out surgical procedures, handling post-surgical pathology, managing seed disposal, and releasing the patient [72].

### 4.5. Costs

The invoicing for the operating room, anesthesia, radiology technicians, and hospital software for material inventory management is the same for RSL and WGL, not including startup costs. When the institution already has a nuclear medicine sector, implementation also becomes more sustainable, as there will be no cost to hire technical staff. Additionally, most materials will already be available. The equipment includes two sodium iodide detectors for seed documentation in the radiology and pathology departments, a navigation system and a portable gamma probe for surgery, and imaging equipment to guide the procedure and for specimen radiography [72].

The cost of materials, as well as the cost per case, which in addition to the materials considers the fees of the professional team and possible complications of the procedure, is similar between the technologies. However, in the long term, considering costs and efficiency in the use of resources, there is a cost saving with RSL per case in both the public and private health services, based on the flexibility of scheduling between marker implantation and surgery, with fewer delay times in the operating room. The longer interval time between RSL localization and surgery results in significant efficiency gains for the radiology team, due to the ability to perform all the week’s procedures in a single list, and for the surgical team, as the patient undergoing RSL can be the first of the day without the delays associated with waiting for the implant to be performed on the same day [50,69].

Assuming a decrease in the reoperation rate with RSL, in the public health service there is a decrease in costs per case, while in the fee-for-service model switching to RSL results in a loss of revenue per case, because without reoperation the number of surgeries decreases.

## 5. Conclusions

This study presented a systematic review and meta-analysis comparing radioactive seed localization (RSL) versus radio-guided occult lesion localization (ROLL) and wire-guided localization (WGL) for patients with impalpable breast cancer undergoing breast-conserving surgery, evaluating the efficacy, safety, and logistical outcomes. Selected studies were submitted to their own data extraction forms and risk of bias analysis according to the ROB 2 and ROBINS 1 tools. The meta-analysis was performed, considering the random effect model, calculating the relative risk or the mean difference for dichotomous or continuous data, respectively. The quality of the evidence generated was analyzed by outcome according to the GRADE tool. Overall, 46 articles met the inclusion criteria and were included in this systematic review, including 4 studies comparing RSL and ROLL with a population of 1550 women and 43 comparing RSL and WGL with a population of 19,820 women. The results showed that RSL is a superior method to WGL in terms of surgical efficiency in the impalpable breast lesions’ intraoperative localization and is at least equivalent to ROLL. Regarding security, RSL obtained results equivalent to the already established technique, the WGL. In addition to presenting promising results about the organization of services, RSL was proven to be superior to WGL and ROLL technologies.

## Figures and Tables

**Figure 1 diagnostics-14-00441-f001:**
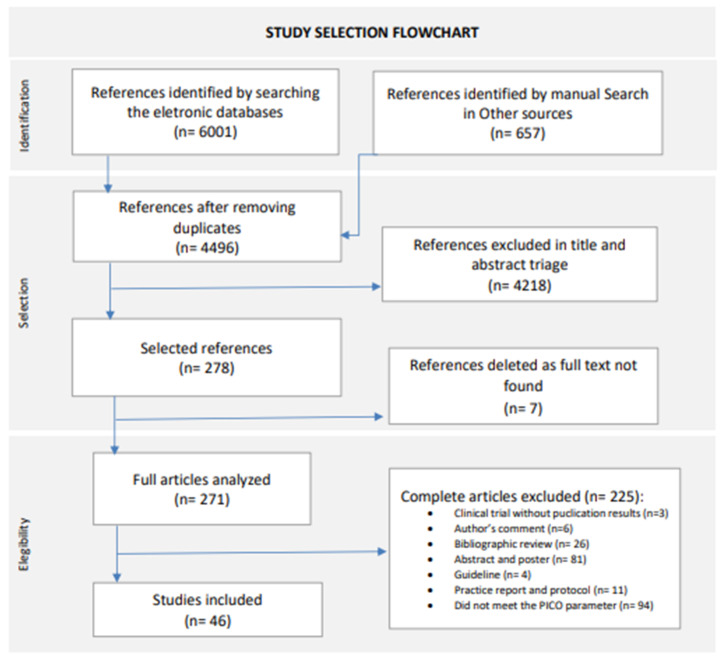
Study selection flowchart. New version adapted from Page, M.J., 2021 [21].

**Figure 2 diagnostics-14-00441-f002:**
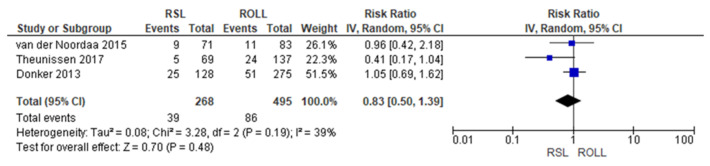
Positive surgical margin forest plot for RSL and ROLL. Figure generated by Review Manager software. References cited are [19,27,29].

**Figure 3 diagnostics-14-00441-f003:**
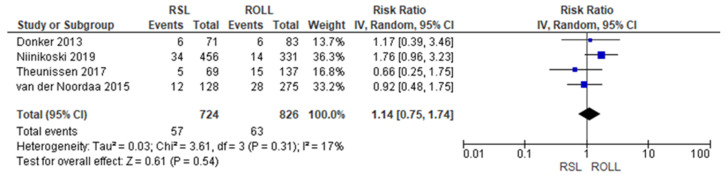
Reoperation forest plot for RSL and ROLL. Figure generated by Review Manager software. References cited are [19,27,28,29].

**Figure 4 diagnostics-14-00441-f004:**
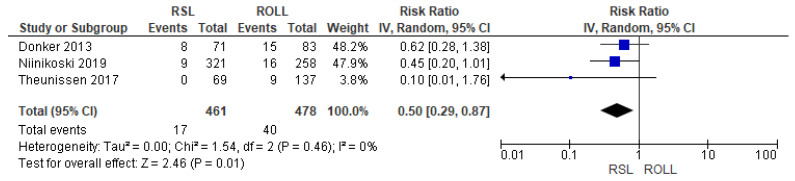
Recurrence forest plot for RSL and ROLL. Figure generated by Review Manager software. References cited are [27,28,29].

**Figure 5 diagnostics-14-00441-f005:**
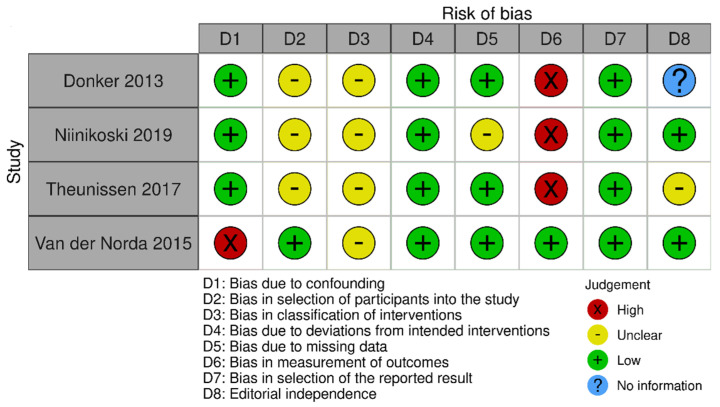
Risk of bias assessment result of non-randomized clinical trials comparing RSL and ROLL. Figure generated by Robvis software (R package, 2019) developed by University of Bristol, UK, 2019. References cited are [19,27,28,29].

**Figure 6 diagnostics-14-00441-f006:**
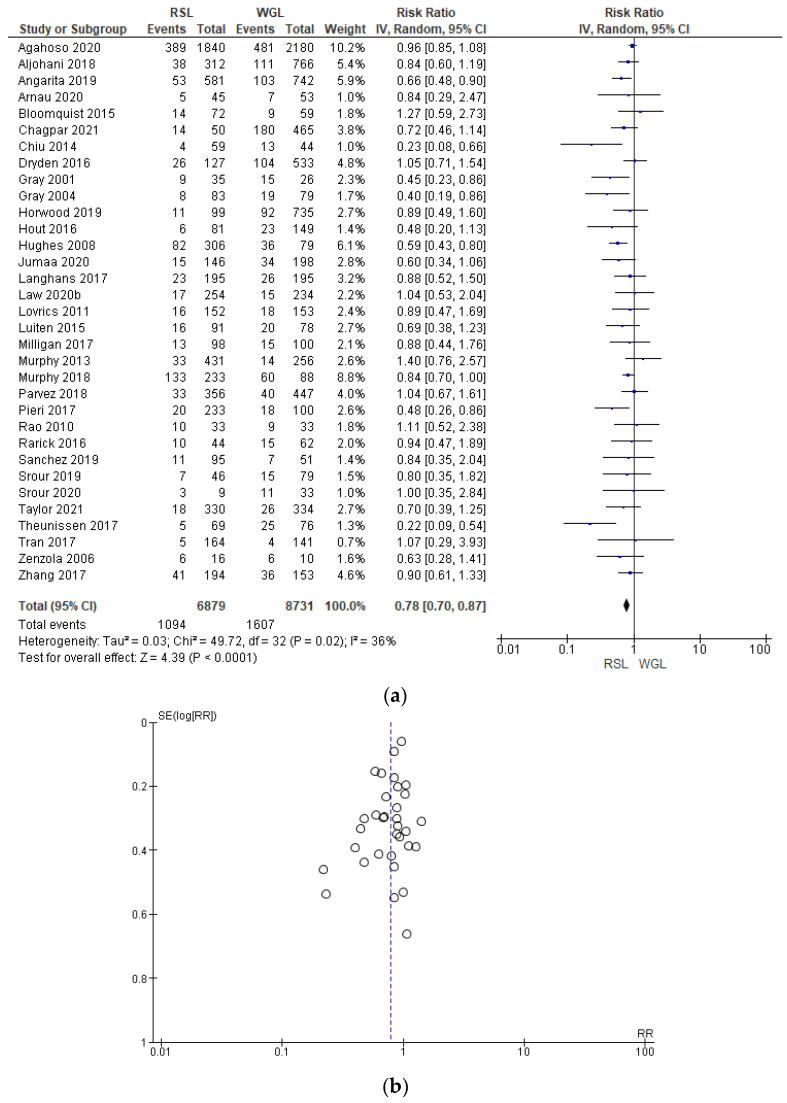
(**a**) Forest plot and (**b**) funnel plot of positive surgical margins with RSL and WGL. Figure generated by the Review Manager software. References cited are [29,30,31,33,34,35,38,39,40,41,42,44,45,46,47,48,49,50,51,52,53,54,55,56,57,58,60,64,65,67,68,69,70]. Caption: SE = standard error; RR = risk ratio.

**Figure 7 diagnostics-14-00441-f007:**
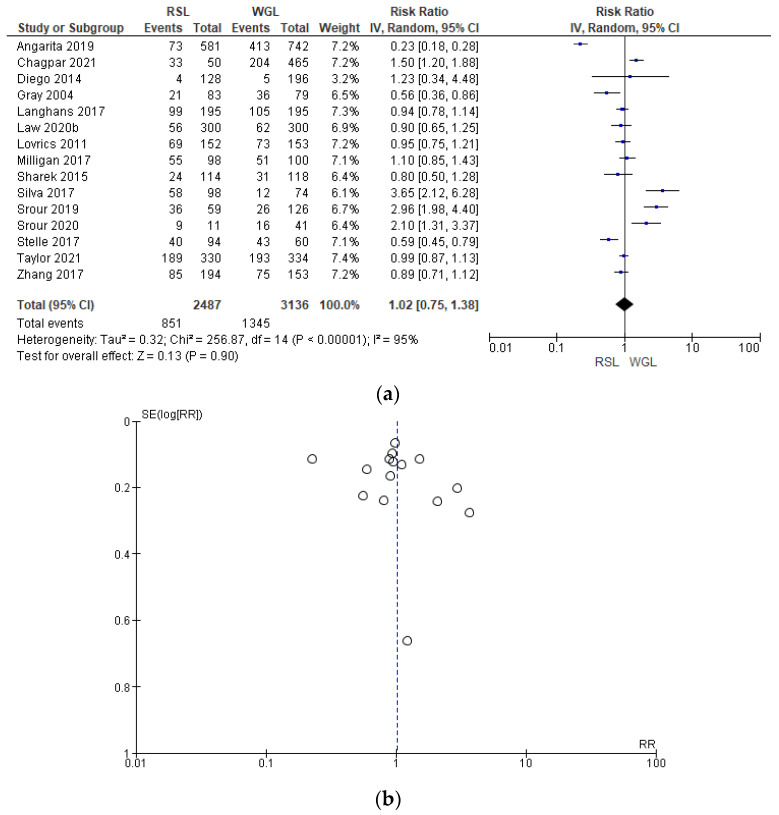
(**a**) Forest plot and (**b**) funnel plot of intraoperative re-excision with RSL and WGL. Figure generated by Review Manager software. References cited are [31,34,35,40,43,45,50,62,63,64,65,66,69,70]. Caption: SE = standard error; RR = risk ratio.

**Figure 8 diagnostics-14-00441-f008:**
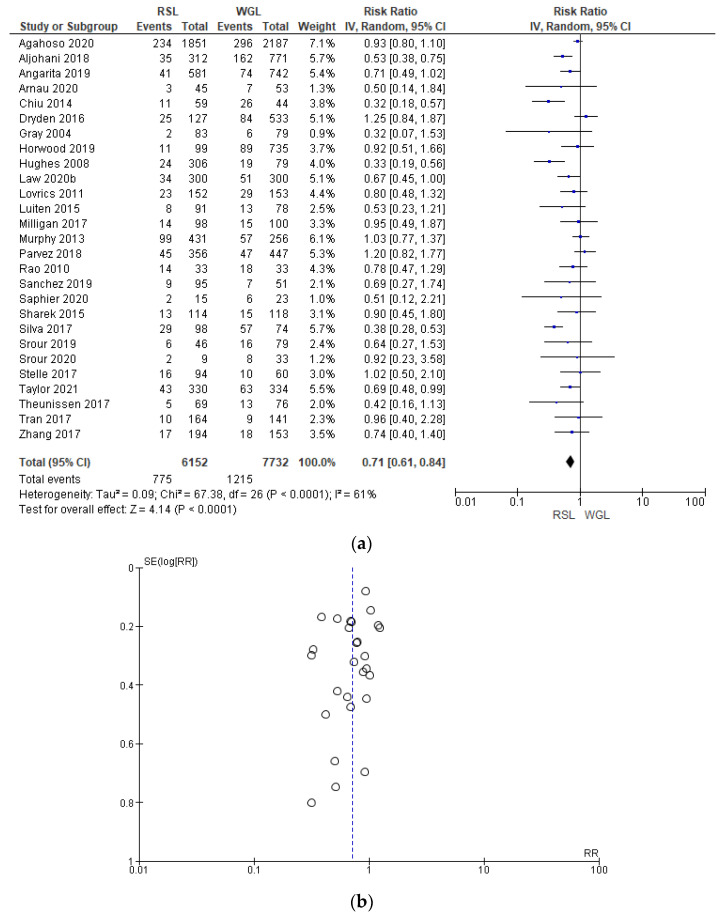
(**a**) Forest plot and (**b**) funnel plot of reoperation with RSL and WGL. Figure generated by Review Manager software. References cited are [29,35,38,39,40,41,42,44,45,46,47,48,49,50,51,52,53,54,55,56,57,60,61,62,63,64,65,66,67,69,70]. Caption: SE = standard error; RR = risk ratio.

**Figure 9 diagnostics-14-00441-f009:**
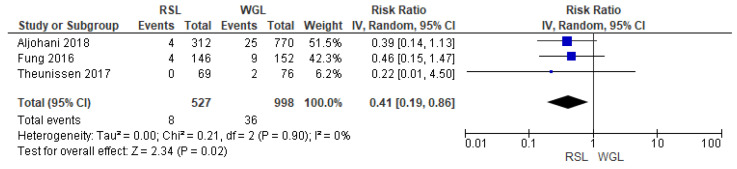
Forest plot of recurrence RSL and WGL. Figure generated by Review Manager software. References cited are [29,32,39].

**Figure 10 diagnostics-14-00441-f010:**
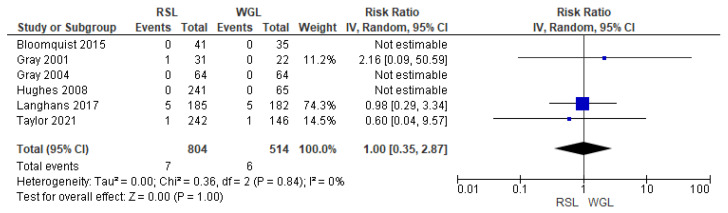
Forest plot of failure identification sentinel lymph node biopsy RSL and WGL. Figure generated by Review Manage software. References cited are [30,33,34,45,48,70].

**Figure 11 diagnostics-14-00441-f011:**
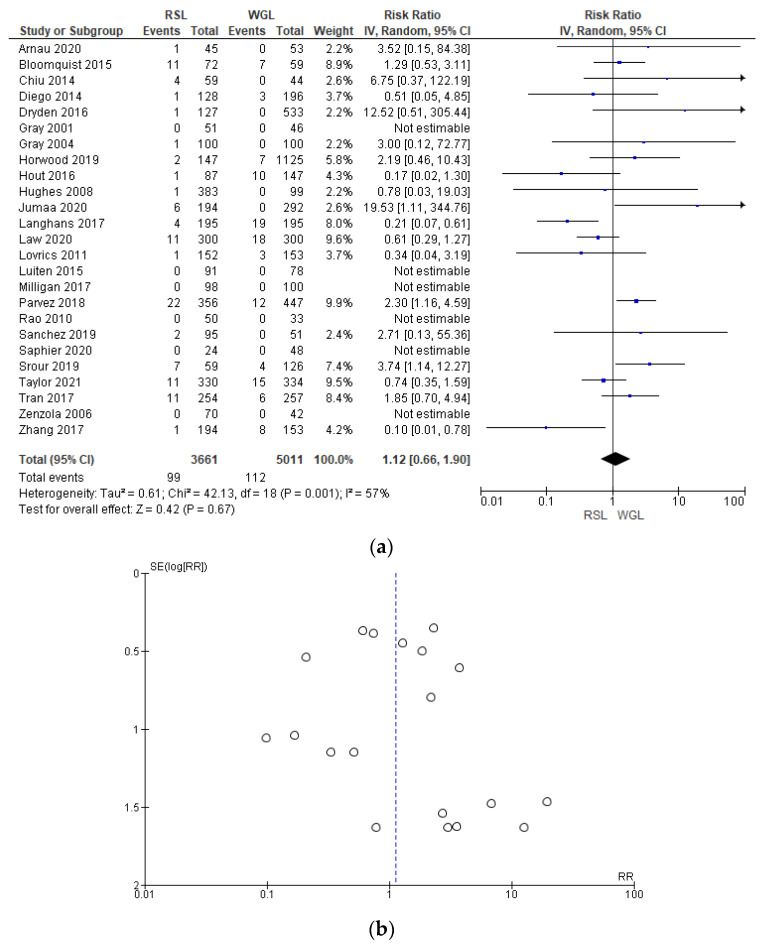
(**a**) Forest plot and (**b**) funnel plot of marker complications with RSL and WGL. Figure generated by Review Manager software. References cited [30,33,34,35,41,42,43,44,45,46,47,48,49,50,51,52,55,57,60,61,64,67,68,69,70]. Caption: SE = standard error; RR = risk ratio.

**Figure 12 diagnostics-14-00441-f012:**
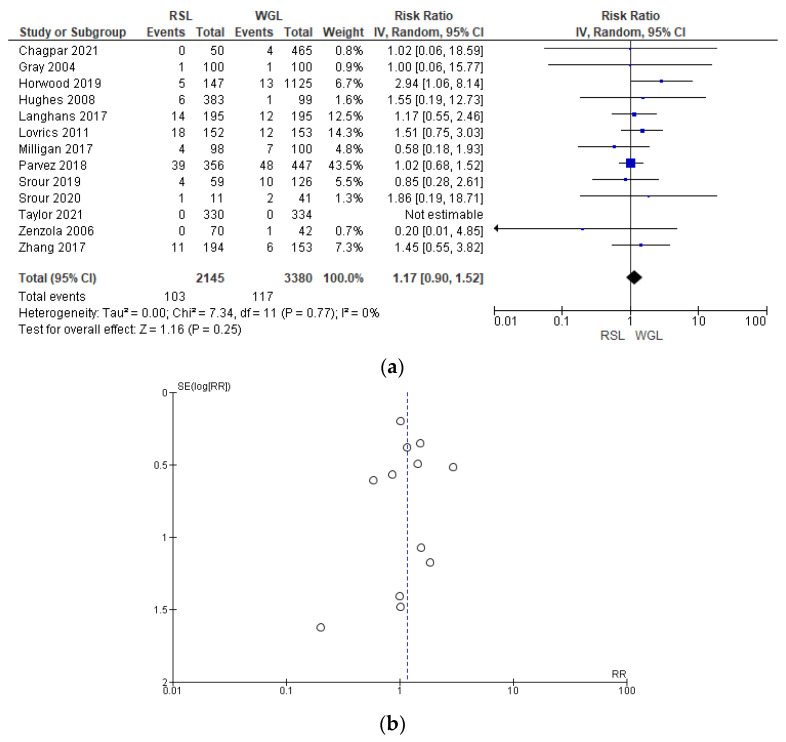
(**a**) Forest plot and (**b**) funnel plot of post-operative complications with RSL and WGL. Figure generated by Review Manager software. References cited [29,31,32,41,42,44,48,51,60,61,64,65,66]. Caption: SE = standard error; RR = risk ratio.

**Figure 13 diagnostics-14-00441-f013:**
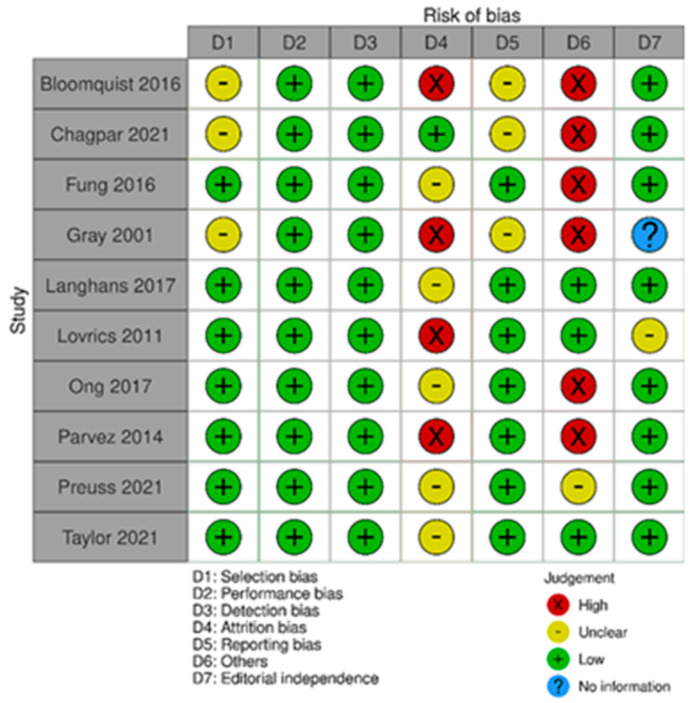
Risk of bias assessment result of randomized clinical trials comparing RSL and WGL. Figure generated by Robvis software. References cited are [30,31,32,33,34,35,36,50,70].

**Figure 14 diagnostics-14-00441-f014:**
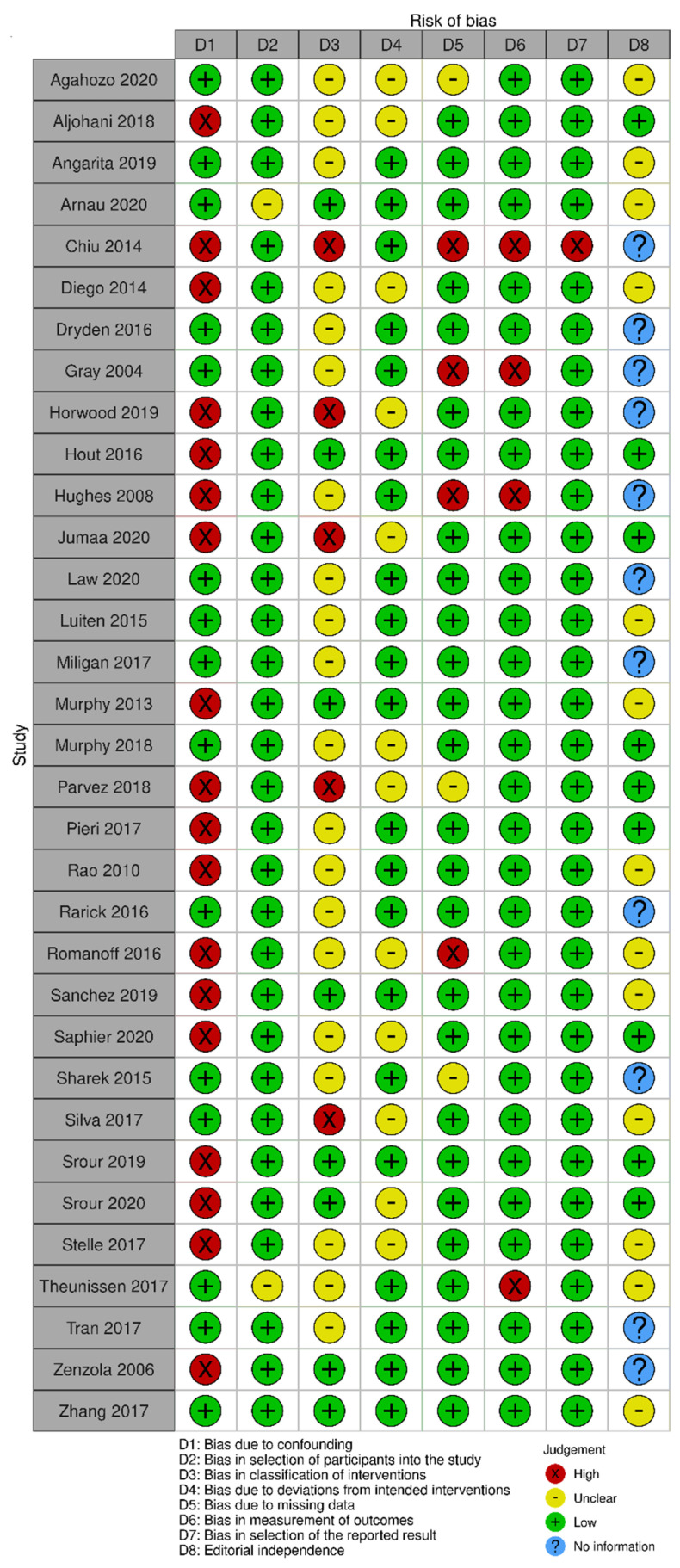
Risk of bias assessment result of non-randomized clinical trials comparing RSL and WGL. Figure generated by Robvis software. References cited are [29,38,39,40,41,42,43,44,45,46,47,48,49,50,51,52,53,54,55,56,57,58,59,60,61,62,63,64,65,66,67,68,69].

**Table 1 diagnostics-14-00441-t001:** Patients’ characteristics in RSL and ROLL groups.

Data	RSL	ROLL
Age	28–91	
Number of participants	724	826
Carcinoma ductal in situ	117	106
Carcinoma invasive	607	720
Neoadjuvant chemotherapy	71	83
Bracketing localization	2	0
Unknown bracketing localization	69	-

**Table 2 diagnostics-14-00441-t002:** Patient’s characteristics in RSL and WGL groups.

Data	RSL	WGL
Number of participants	8670	11,150
Age 20–92 years old	6986	7990
Unknown age	1684	3160
Carcinoma ductal in situ	2917	3593
Carcinoma invasive	4432	4481
Other pathologies ^1^	1321	3076
Neoadjuvant chemotherapy	161	77
Unknown neoadjuvant chemotherapy	418	839
Bracketing localization	479	577
Unknown bracketing localization	1059	2579
Excisional biopsy	112	177
Unknown excisional biopsy	1607	2556

^1^ Includes other types of carcinomas, high-risk lesions, benign lesions, and cases of unknown pathology.

## Data Availability

No new data were created.

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
