# Peer review of "Radioactive Seed Localization for Nonpalpable Breast Lesions: Systematic Review and Meta-Analysis"

_diagnostics, 2024, doi:10.3390/diagnostics14040441_

Round 1

Reviewer 1 Report

Comments and Suggestions for Authors

see attached file

Author Response

Dear Editor

We have incorporated all of your suggestions. We appreciate your time and efforts and wish to sincerely thank you.

Reviewer 2 Report

Comments and Suggestions for Authors

Dear Respected Authors,

Thank you for submitting your manuscript entitled "Radioactive seed localization for nonpalpable breast lesions: systematic review and meta-analysis." Your work represents a valuable contribution to the field, addressing an important topic in breast cancer surgery. The systematic review and meta-analysis comparing radioactive seed localization (RSL) with radio-guided occult lesion localization (ROLL) and wire-guided localization (WGL) for patients with impalpable breast cancer undergoing breast-conserving surgery are of significant interest.

However, I would like to provide some constructive feedback to enhance the clarity and impact of your manuscript. Firstly, while the results are promising, they appear to be overly extended and need to be summarized and structured more effectively. Additionally, I suggest a more thorough discussion of the reached results, as this could significantly enhance the value of the manuscript by providing insights and context.

I have identified some minor comments for improvement:

- Consider not separating the different sections in the introduction.

- Reframe the systematic review section into the study objective using past tense.

- Clearly note the inclusion and exclusion criteria in the body of the manuscript instead of supplementary material.

- Justify the exclusion of Scopus as one of the databases in your search.

- Explain abbreviations in the footnote of the tables.

- Summarize supporting data for forest and funnel plots next to the plots in an editable version.

- Update references with recent publications (after 2022).

Major comments:

- Rewrite the methods section, avoiding a copy-paste of the journal guideline, and craft your own methods using the platform.

- Revise the abstract to ensure it is not a direct copy-paste of the journal guideline but a genuine representation of the study.

- Ensure all supplementary materials are accessible.

- Address the wide age range of included patients and explore potential differences among age groups.

- Extend Figure 1 to show clearly how many studies were excluded from the meta-analyses due to a high risk of bias.

- Expand the "Comparison with similar researches" section in the discussion, delving into interesting results found in the included manuscripts.

- Provide clear recommendations for future studies, moving beyond a single sentence in the limitations section.

- Discuss the high risk of bias among included studies in a separate paragraph within the discussion.

- Expand the "Implications and actions needed" section, providing insights on implementing results into clinical practice.

- Discuss the costs of the two methods, considering the potential implications for different healthcare systems.

Author Response

Dear R2

Please see the attached file. We appreciate your time and efforts and wish to sincerely thank you.

Round 2

Reviewer 2 Report

Comments and Suggestions for Authors Upon reviewing the author's revisions in the journal's system, I am pleased to note that their responses have been comprehensive, and the revisions made to the manuscript appear to be satisfactory. In my opinion, the manuscript is now suitable for acceptance. Comments on the Quality of English Language Upon reviewing the author's revisions in the journal's system, I am pleased to note that their responses have been comprehensive, and the revisions made to the manuscript appear to be satisfactory. In my opinion, the manuscript is now suitable for acceptance.